# Two-Neuron Based Memristive Hopfield Neural Network with Synaptic Crosstalk

**Rong Qiu** [1], **Yujiao Dong** [2,*], **Xin Jiang** [2] **and Guangyi Wang** [2]

1   School of Internet of Things Engineering, Guangdong Polytechnic of Science and Technology, Guangzhou 510640, China
2   Institute of Modern Circuits and Intelligent Information, Hangzhou Dianzi University, Hangzhou 310018, China
*   Correspondence: yjdong@hdu.edu.cn

**Abstract:** Synaptic crosstalk is an important biological phenomenon that widely exists in neural networks. The crosstalk can influence the ability of neurons to control the synaptic weights, thereby causing rich dynamics of neural networks. Based on the crosstalk between synapses, this paper presents a novel two-neuron based memristive Hopfield neural network with a hyperbolic memristor emulating synaptic crosstalk. The dynamics of the neural networks with varying memristive parameters and crosstalk weights are analyzed via the phase portraits, time-domain waveforms, bifurcation diagrams, and basin of attraction. Complex phenomena, especially coexisting dynamics, chaos and transient chaos emerge in the neural network. Finally, the circuit simulation results verify the effectiveness of theoretical analyses and mathematical simulation and further illustrate the feasibility of the two-neuron based memristive Hopfield neural network hardware.

**Keywords:** memristor; Hopfield neural network; chaos; synaptic crosstalk; coexisting dynamics

## 1. Introduction

Neural systems contribute to processing information in brains, where neurons and synapses play an important role in transmitting information. It is reported that crosstalk exists between synapses because of their interaction [1]. When the neurotransmitter overflow between synapses or the diffusion of receptors between neighboring ridges emerges, the signal transmission may be affected, thereby inducing the variation in the functions of brains.

Memristor, defined by Chua in 1971 and physically realized by HP lab in 2008, can be used to emulate synaptic functions [2,3]. By adjusting the voltage across the memristor, the change in memristances can emulate the plasticity of synaptic weights, showing the activity-dependent change in neuronal connection strength. Thus, a memristor is a good choice to replace the fixed resistor-based synapses in the neural networks, which can flexibly solve different kinds of combinatorial optimization problems, such as MAX-CUT problems and TSP (Traveling Salesman Problem) [4].

Recently, memristive neural networks have attracted more and more attention [5–7]. The memristor-based cellular neural network was applied to image processing, which has nonvolatility, compactness, and programmability of synaptic weights [5]. The memristor-based pulse coupled neural network was designed to solve image fusion problems, which improves the quality of images [6]. Ma et al. established a memristor-based Hopfield neural network and emulated human emotions via the circuit simulation [7].

The Hopfield neural network (HNN) proposed by Hopfield is a well-known and typical artificial neural network [8], which has the ability to emulate complex dynamics of the human brain, such as chaos. After that, the HNN is widely applied in associative memory, image processing, combinatorial optimization, and so on [9–13]. Reference [9] proposed a novel algorithm called Teamwork Optimization Algorithm (TOA) to solve

the optimization problems. Kasihmuddin et al. presented an integrated representation of *k*-satisfiability (*k*SAT) in a mutation HNN (MHNN), which overcomes the overfitting issue [10]. Citko et al. set up associative memories to retrieve images based on the HNN [11]. Reference [12] embedded the logical rule $P_{RAN3SAT}$ in HNN and optimized the ability of retrieval. Rubio-Manzano et al. created a complete explainer video about the HNN on a recognition problem [13].

Considering the great advantages of memristors, researchers established Hopfield neural networks based on memristors [14–18]. Sun et al. achieved the recognition and sequence of four characters [16]. Reference [17] explored nonlinear dynamics of a three-neuron based memristive HNN.

This paper aims to study more complex nonlinear behaviors via a simple two-neuron based HNN. In this paper, a two-neuron based memristive HNN with synaptic crosstalk is established. By analyzing the dissipation and stability of the HNN, many different types of coexisting dynamics are found. It is verified that the memristive parameter and crosstalk weight have a significant influence on the complexity of the HNN via the bifurcation diagram, basin of attraction, and so on. Finally, the circuit simulation results verify the effectiveness of theoretical analyses.

## 2. Simplest Hyperbolic Memristive Synapse-Coupled HNN

### 2.1. Hyperbolic Memristive Synapse Emulator

Neuron activation functions are used to transform the output signal of the former neuron into the input signal of the latter neuron, where the sigmoid nonlinear activation function is commonly used.

Here, we use the hyperbolic tangent function with zero-mean as the activation function, which has a higher range and greater slope than the sigmoid activation function. The mathematical description of the hyperbolic tangent function is

$$v_o = -\tanh(v_i) \tag{1}$$

The equivalent circuit of the inverting hyperbolic tangent function is shown in Figure 1, including two operational amplifiers TL084 ($U_i$ and $U_o$), two transistors MPS2222 ($Q_1$ and $Q_2$), one current source $I_0 = 1.1$ mA, and several resistors ($R_1 = R_5 = R_6 = R_7 = R_8 = 10$ k$\Omega$, $R_2 = 0.52$ k$\Omega$, $R_3 = R_4 = 1$ k$\Omega$), where $V_{in}$ and $V_{out}$ are the input and output voltage, respectively. The operational amplifiers $U_i$ and $U_o$ finished the inversion of the input and subtraction operation, respectively. The transistors $Q_1$ and $Q_2$ realized the exponential operation.

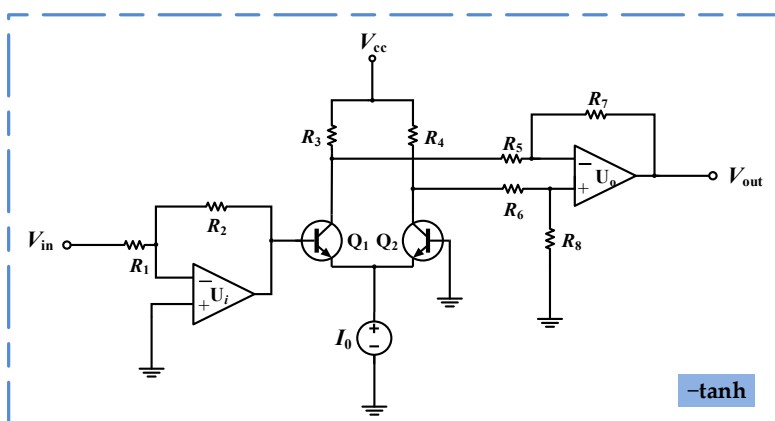

**Figure 1.** Circuit scheme of inverting hyperbolic tangent function.

The obtained simulated results of the inverting hyperbolic tangent circuit using Pspice are shown in Figure 2.

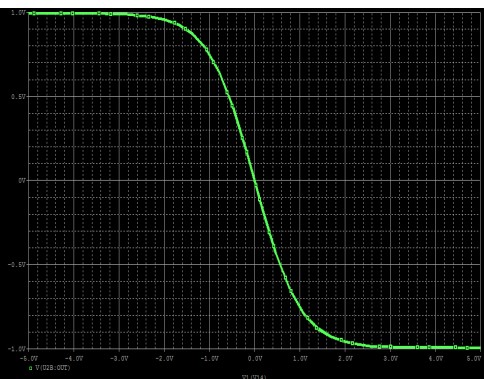

**Figure 2.** Simulated results of the inverting hyperbolic tangent circuit.

The generic model of hyperbolic tangent-type memristor emulator can be used to emulate synaptic weights of neurons, which is described as

$$i = W(x)v = [a - b\tanh(x)]v \tag{2}$$

where $v$, $i$, and $x$ represent the voltage, current, and state variable of the memristor; $a$ and $b$ are the memristor parameters, $a > 0$, $b > 0$.

Based on Equation (2), the circuit equation of the memristor can be defined as

$$i = W(x)v = [-\frac{1}{R_a} - \frac{1}{R_b}\tanh(x)]v \tag{3}$$

From Equation (3), one can establish the hyperbolic tangent memristor emulator, as depicted in Figure 3, including an operational amplifier TL084 ($U_o$), a capacitor ($C = 100$ nF), four resistors ($R = 10$ kΩ, $R_a$ and $R_b$ are adjustable), a multiplier AD633, and a model of -tanh.

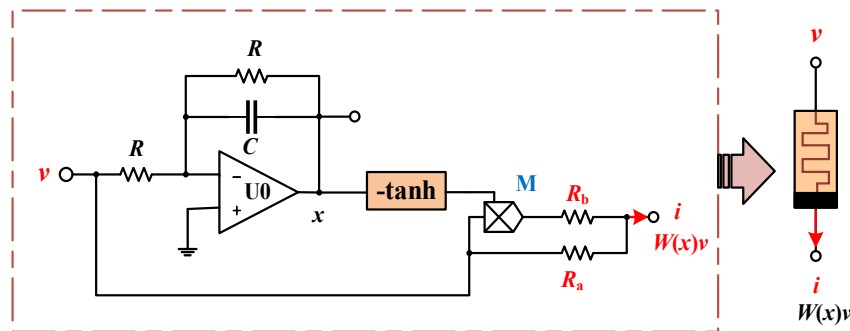

**Figure 3.** Hyperbolic tangent memristor emulator.

A coupled memristor emulator is obtained by coupling the two same hyperbolic tangent memristors [19], as shown in Figure 4, which can emulate the crosstalk between synapses. Thus, the synaptic weights between neurons can be described as

$$\begin{cases} W_1 = a_1 - b_1\tanh(z) + c_1\tanh(u) \\ W_2 = a_2 - b_2\tanh(u) + c_2\tanh(z) \end{cases} \tag{4}$$

where $a_1$, $b_1$, $a_2$ and $b_2$ are memristor parameters; $c_1$ and $c_2$ are crosstalk strength parameters; $a_1 = \frac{R}{R_{a1}}$, $b_1 = \frac{R}{R_{b1}}$, $c_1 = \frac{gR^2}{R_{b1}R_{c1}}$, $a_2 = \frac{R}{R_{a2}}$, $b_2 = \frac{R}{R_{b2}}$, $c_2 = \frac{gR^2}{R_{b2}R_{c2}}$.

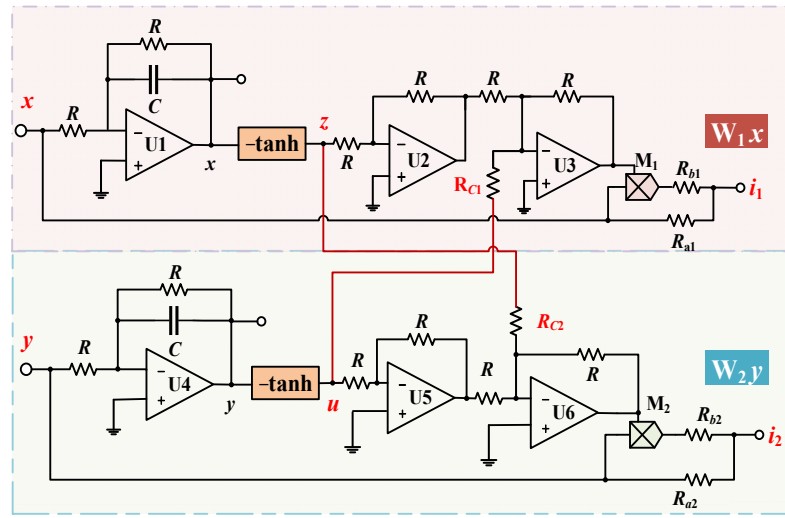

**Figure 4.** Hyperbolic-type memristor emulator with crosstalk.

### 2.2. Two Neurons-Based HNN Model

The Hopfield neural network (HNN), a fully interconnected neural network, can be used to describe dynamic behaviors of human brains [20]. An *n*-neuron-based HNN is defined as

$$C_i \frac{dx_i}{dt} = -\frac{x_i}{R_i} + \sum_{i=1}^{n} w_{ij}\tanh(x_i) + I_i \tag{5}$$

where $C_i$, $R_i$ and $x_i$ represent membrane capacitance, membrane resistance and voltage; $\tanh(x_i)$ is the activation function of neuron *I*; $w_{ij}$ is the synaptic weight between neurons *i* and *j*; $I_i$ is the biasing current.

Here, we design a novel two-neuron-based memristive HNN, as depicted in Figure 5, which can emulate synaptic crosstalk. The mathematical description of the established HNN is

$$\begin{cases} \dot{x} = -x + w_{11}\tanh(x) - k_2 W_2 \tanh(y) \\ \dot{y} = -y + k_1 W_1 \tanh(x) + w_{22}\tanh(y) \\ \dot{z} = -z + \tanh(x) \\ \dot{u} = -u + \tanh(y) \end{cases} \tag{6}$$

where $k_1 = 1, k_2 = 1; w_{11}$ and $w_{22}$ are self-connected synaptic weights; $W_1 = a_1 - b_1\tanh(z) + c_1\tanh(u)$ and $W_2 = a_2 - b_2\tanh(u) + c_2\tanh(z)$ are mutual synaptic weights between neuron 1 and neuron 2. The synaptic weight matrix is

$$W_{ij} = \begin{pmatrix} w_{11} & w_{12} \\ w_{21} & w_{22} \end{pmatrix} = \begin{pmatrix} w_{11} & -k_2 W_2 \\ k_1 W_1 & w_{22} \end{pmatrix} \tag{7}$$

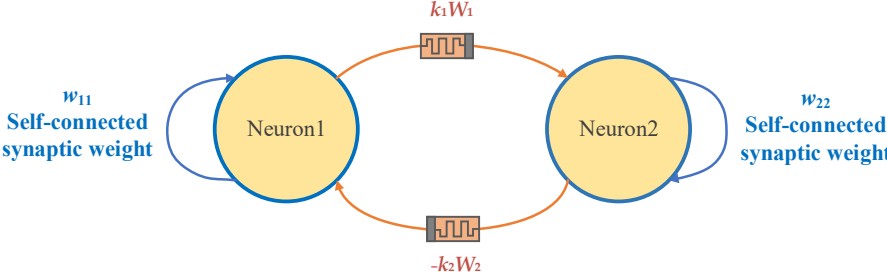

**Figure 5.** The simplest memristive HNN with synaptic crosstalk.

## 3. Dissipativity and Stability of HNN

### 3.1. Dissipativity Analyses

A dissipative characteristic is necessary for a system or network to generate chaos. Thus, the volumetric shrinkage rate $\Lambda$ should be calculated to verify that the system in Equation (6) is dissipative based on $V(t) = Ve^{\Lambda t}$. If $\Lambda < 0$, the system is dissipative, and chaos may emerge; when $\Lambda = 0$, the system is called a conservative system; the divergent phenomenon will occur in the system when $\Lambda > 0$.

According to the method in reference [21], the Lyapunov function is introduced as

$$V(x, y, z, u) = \frac{1}{2}\left(x^2 + y^2 + z^2 + u^2\right) \tag{8}$$

whose corresponding time derivative is

$$
\begin{aligned}
\dot{V}(x, y, z, u) &= x\dot{x} + y\dot{y} + z\dot{z} + u\dot{u} \\
&= -\left(x^2 + y^2 + z^2 + u^2\right) + v(x, y, z, u) \\
&= -2V(x, y, z, u) + v(x, y, z, u)
\end{aligned}
\tag{9}
$$

where

$$v(x, y, z, u) = (w_{11}x + W_1x + z)\tanh(x) + (w_{22}y + W_2x + u)\tanh(y) \tag{10}$$

Since $\tanh(\zeta) \in (-1, 1)$ for all $\zeta = x, y, u, z$, one can obtain

$$
\begin{aligned}
|W_1| &= a_1 - b_1\tanh(z) + c_1\tanh(u) \\
&\leq M_1 = \max\{|a_1 - b_1 + c_1|, |a_1 + b_1 + c_1|\} \\
|W_2| &= a_2 - b_2\tanh(u) + c_2\tanh(z) \\
&\leq M_2 = \max\{|a_2 - b_2 + c_2|, |a_2 + b_2 + c_2|\}
\end{aligned}
\tag{11}
$$

So we have

$$
\begin{aligned}
v(x, y, z, u) &\leq |(w_{11}x + W_1y + z)\tanh(x)| + |(w_{22}y + W_2x + u)\tanh(y)| \\
&\leq (w_{11} + M_2)|x| + (M_1 + w_{22})|y| + |z| + |u|
\end{aligned}
\tag{12}
$$

Suppose that all state variables $(x, y, z, u)$ satisfy $V(x, y, z, u) = D$ for $D > D_0$ ($D_0 > 0$ is a sufficiently large domain), it requires

$$
\begin{aligned}
v(x, y, z, u) &< (w_{11} + M_2)|x| + (M_1 + w_{22})|y| + |z| + |u| \\
&< x^2 + y^2 + z^2 + u^2 = 2V(x, y, z, u)
\end{aligned}
\tag{13}
$$

where $w_{11} + M_2$ and $M_1 + w_{22}$ are positive constants. So

$$\{(x, y, z, u)|V(x, y, z, u) = D\} \tag{14}$$

Since $D > D_0$, one can obtain

$$\dot{V} = -2V(x, y, z, u) + v(x, y, z, u) < 0 \tag{15}$$

Based on Equation (15), the confined domain of the solutions in Equation (6) is given, as

$$\{(x, y, z, u)|V(x, y, z, u) \leq D\} \tag{16}$$

Thus, the memristive HNN in Equation (6) is bounded, which has the possibility to generate chaos.

### 3.2. Stability Analyses

From Equation (6), the equilibria $P = (\overline{x}, \overline{y}, \overline{z}, \overline{u}, \overline{w})$ can be calculated by

$$\begin{cases} 0 = -\overline{x} + w_{11}\tanh(\overline{x}) - k_2 W_2\tanh(\overline{y}) \\ 0 = -\overline{y} + k_1 W_1\tanh(\overline{x}) + w_{22}\tanh(\overline{y}) \\ 0 = -\overline{z} + \tanh(\overline{x}) \\ 0 = -\overline{u} + \tanh(\overline{y}) \end{cases} \tag{17}$$

where $W_1$ and $W_2$ are hyperbolic-type memristor emulator, as

$$\begin{cases} W_1 = a_1 - b_1\tanh(z) + c_1\tanh(u) \\ W_2 = a_2 - b_2\tanh(u) + c_2\tanh(z) \end{cases} \tag{18}$$

By solving Equations (17) and (18), one can obtain

$$\begin{cases} H_1(x, y) = -x + w_{11}\tanh(x) - k_2(a_2 - b_2\tanh(u) + c_2\tanh(z))\tanh(y) \\ H_2(x, y) = -y + k_1(a_1 - b_1\tanh(z) + c_1\tanh(u))\tanh(x) + w_{22}\tanh(y) \end{cases} \tag{19}$$

where the equilibria of the HNN are the intersection points between the curve $H_1(x, y)$ and $H_2(x, y)$.

Now, as an example, we set $a_1 = 1$, $a_2 = 2$, $b_1 = 0.04$, $b_2 = 0.03$, $c_1 = 5.55$ and $c_2 = 5.9$. The two curves $H_1(x, y)$ and $H_2(x, y)$ are shown in Figure 6. Based on Figure 6 and Equation (17), the obtained equilibria are $P_0$ (0, 0, 0, 0), $P_1$ (−1.7, −0.238, −0.935, 0.234), $P_2$ (−0.174, −1.116, −0.184, −0.806) and $P_3$ (1.737, −0.149, 0.940, −0.148).

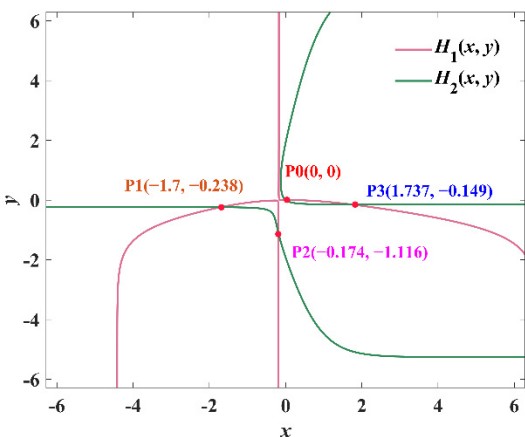

**Figure 6.** Equilibria of the memristive HNN, i.e., the intersection points of $H_1$ and $H_2$.

By linearizing Equation (6) at the equilibria $(\overline{x}, \overline{y}, \overline{z}, \overline{u})$, one obtains its Jacobian matrix as

$$J = \begin{bmatrix} -1 + w_{11}h_1 & -W_2 h_2 & -c_2\tanh(\overline{y})h_3 & b_2\tanh(\overline{y})h_4 \\ W_1 h_1 & -1 + w_{22}h_2 & -b_1\tanh(\overline{x})h_3 & c_1\tanh(\overline{x})h_4 \\ h_1 & 0 & -1 & 0 \\ 0 & h_2 & 0 & -1 \end{bmatrix} \tag{20}$$

where $h_1 = 1 - \tanh(\overline{x})$, $h_2 = 1 - \tanh(\overline{y})$, $h_3 = 1 - \tanh(\overline{z})$, $h_4 = 1 - \tanh(\overline{u})$.

According to the stability criterion, at least one positive eigenvalue causes an unstable equilibrium. The eigenvalues at different equilibria and their stability are listed in Table 1.

**Table 1.** Equilibria, eigenvalues and their stability of the HNN.

| Equilibria | Eigenvalues | Stability |
|---|---|---|
| $P_0$ (0, 0, 0, 0) | $0.5000 \pm 0.8944i$, $-1.0000$, $-1.0000$ | unstable |
| $P_1$ $(-1.7, -0.238, -0.935, 0.234)$ | $-0.0760 \pm 1.6875i$, $-1.2437$, $-0.5887$ | stable |
| $P_2$ $(-0.186, -1.116, -0.184, -0.806)$ | $1.6461$, $-0.6511 \pm 0.2767i$, $-2.6779$ | unstable |
| $P_3$ $(1.737, -0.149, 0.940, -0.148)$ | $2.3973$, $-2.4408$, $-1.1678$, $-0.7158$ | unstable |

### 3.3. Chaotic Behaviors

Here, set the initial condition $(x_0, y_0, z_0, u_0) = (0.1, 0, 0, 0)$, $k_1 = 1$, $k_2 = 1$, $W_{11} = 1$, $W_{22} = 2$, $a_1 = 1$, $a_2 = 1$, $b_1 = 0.04$, $b_2 = 0.03$. The HNN system generates chaos shown in Figure 7. Figure 7a–d show chaotic attractors on the *x-y* phase, *x-z* phase, *x-u* phase and *y-z* phase, respectively. Then, we use the Poincaré map and Lyapunov exponent to verify the chaotic behaviors.

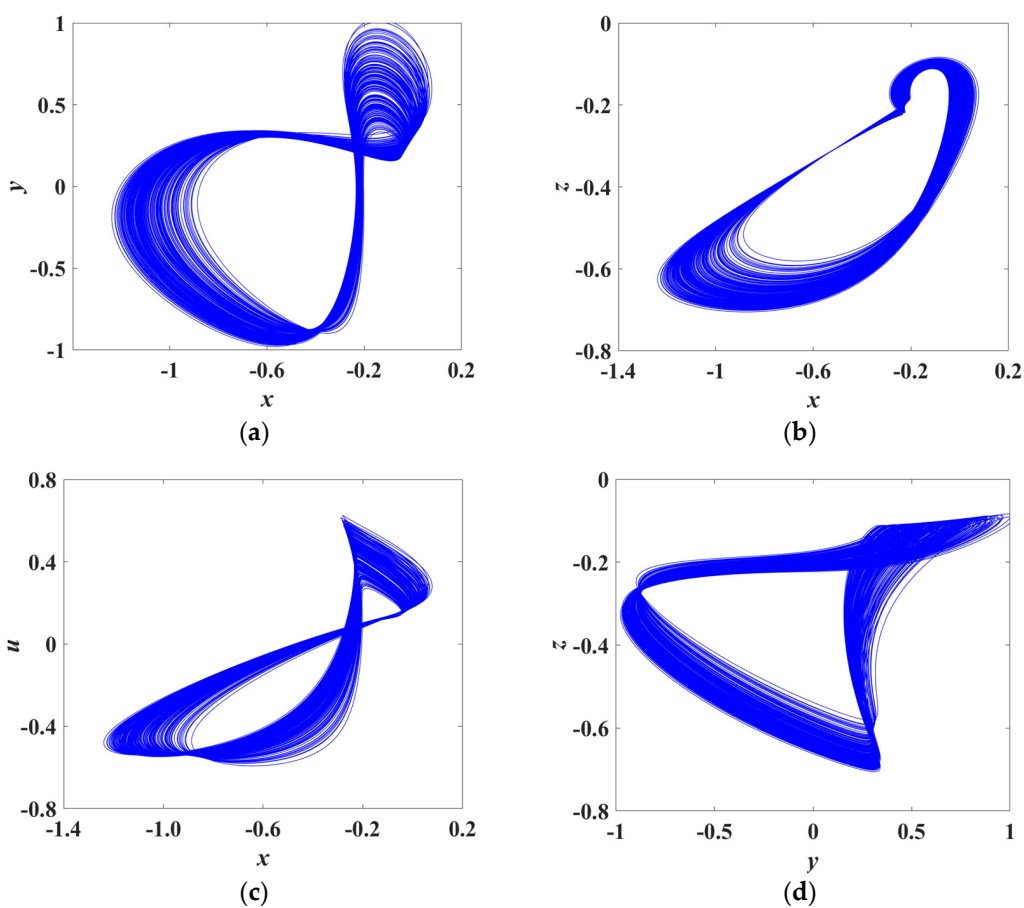

**Figure 7.** Chaotic attractors: (**a**) *x-y* phase portraits; (**b**) *x-z* phase portraits; (**c**) *x-u* phase portraits; (**d**) *y-z* phase portraits.

The Poincaré map is a qualitative method to verify chaotic phenomena. If there are one or several points on the Poincaré map, the system shows stable or periodic characteristics; a large number of dense points in the Poincaré map predict chaos. Figure 8 shows the Poincaré map when the cross-section is chosen as the plane $z = -0.1$. It is found that two continuous segments with dense points emerge on the *x-y* plane, indicating chaotic attractors.

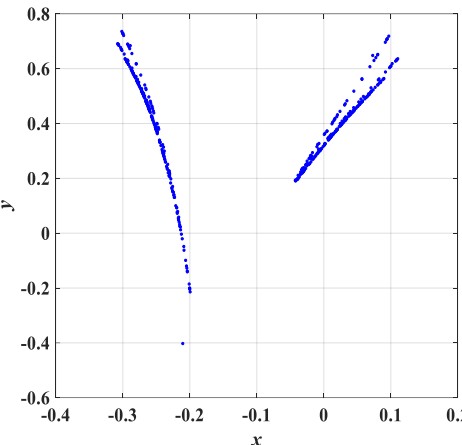

**Figure 8.** Poincaré map in the *x*-*y*-*z* space with the cross-section $z_0 = -0.1$.

The Lyapunov exponent is a quantitative method to judge chaotic attractors. By using the QR decomposition method to calculate the Jacobian matrix and its eigenvalues, the Lyapunov exponents of the system are calculated as $LE_1 = 0.058$, $LE_2 = 4.892 \times 10^{-4}$, $LE_3 = -0.179$, $LE_4 = -1.402$, and the corresponding Lyapunov dimension is $DL = 2.321$. Since the maximum Lyapunov exponent $LE_1 > 0$, the Hyperbolic-type memristive HNN produces chaos.

## 4. Dynamics Varying with Parameters

In this section, we choose two representative parameters $a_2$ and $c_2$ to study the influence on dynamics of the memristive HNN, where $a_2$ and $c_2$ are the memristive parameter and crosstalk parameter, respectively. We use the bifurcation diagram, Lyapunov exponent spectrum, and phase portraits to further explore the complex dynamics of two-neuron based HNNs varying with $a_2$ and $c_2$.

### 4.1. Influence of Memristive Parameter $a_2$ on Dynamics

The synaptic plasticity of neurons can be realized by adding memristors into neural networks, which is important for the HNN to solve many different kinds of combinatorial optimization problems including MAX-CUT problems, TSP, and so on. Changing the memristive parameter means adjusting the synaptic weight. Now, we set the initial condition $(x_0, y_0, z_0, u_0) = (0.1, 0, 0, 0)$, $W_{11} = 1.24$, $W_{22} = 0.75$, $a_1 = 1$, $b_1 = 0.03$, $b_2 = 0.02$, $c_1 = 5.7$, $c_2 = 5.9$. The bifurcation diagram of the HNN and corresponding Lyapunov exponent spectrum varying with $a_2$ over the range of [1.1, 1.5] are shown in Figure 9.

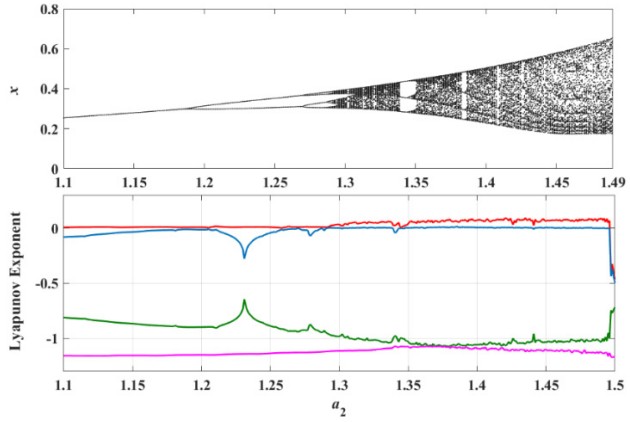

**Figure 9.** The bifurcation diagram and Lyapunov exponent spectrum of the HNN varying with $a_2 \in [1.1, 1.5]$.

Observe from Figure 9 that the memristive parameter $a_2$ has a great influence on the dynamics of the HNN. The chaotic and periodic region in the bifurcation diagram is almost consistent with that of the Lyapunov exponent spectrum. When $a_2 \in [1.1, 1.29]$, the HNN evolves from the single-period state into chaos via the period-doubling bifurcation. As $a_2$ gradually increases to 1.34, the HNN turns into the three-period state and then enters into the chaotic region. When $a_2 \in [1.34, 1.49]$, the HNN is mostly chaotic except for several narrow periodic windows. Interestingly, the HNN exhibits transient chaos when $a_2 = 1.5$ and finally shows non-chaotic phenomena.

The representative phase portraits of the HNN under the parameter $a_2 = 1.1, 1.2, 1.3,$ 1.4 and 1.5 are depicted in Figure 10, corresponding to single-period (red), double-period (blue), quasi-period (green), chaos (purple) and transient chaos (pink), respectively.

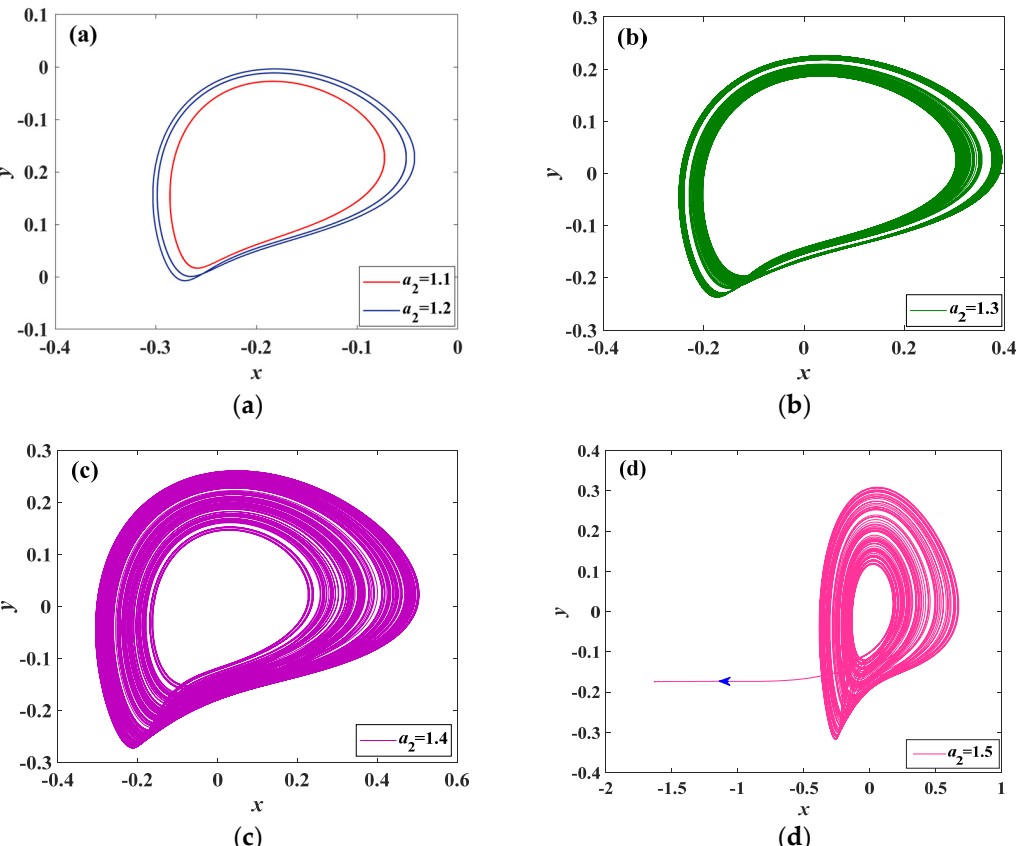

**Figure 10.** Phase portraits of the HNN under the parameter $a_2 = 1.1, 1.2, 1.3, 1.4, 1.5$: (**a**) $a_2 = 1.1, 1.2$; (**b**) $a_2 = 1.3$; (**c**) $a_2 = 1.4$; (**d**) $a_2 = 1.5$.

The time-domain waveforms under $a_2 = 1.4$ and $a_2 = 1.5$ are depicted in Figure 11, corresponding to red and green trajectories. Observe that the HNN shows chaotic states over the range of $t \in [200 \text{ s}, 550 \text{ s}]$ with $a_2 = 1.4$ and $a_2 = 1.5$. However, the HNN evolves from the chaotic state into the stable state over the range of $t \in [550 \text{ s}, 800 \text{ s}]$ when $a_2 = 1.5$. This special phenomenon is called transient chaos [22,23], with short-time chaotic behaviors.

Thus, changing memristive parameter $a_2$ can adjust synaptic weights, and finally, the dynamics of the HNN are easily controlled.

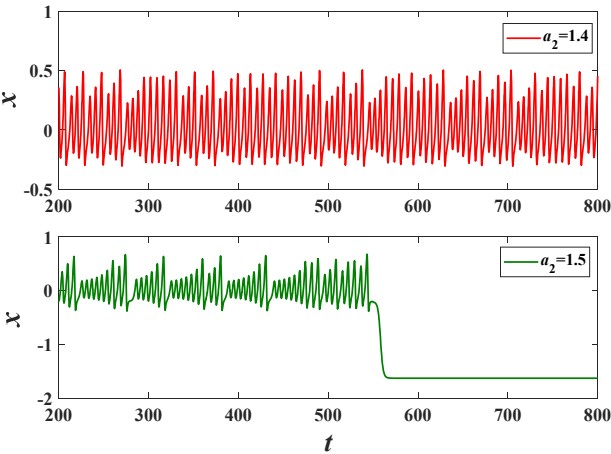

**Figure 11.** Time-domain waveforms with $a_2$ = 1.4 (red) and 1.5 (green).

### 4.2. Influence of Crosstalk Parameter $c_2$ on Dynamics

Here, set the parameter $c_1$ = 5.56. The bifurcation diagram of the HNN over the range of $c_2 \in$ [5.5, 6] is shown in Figure 12. Observe that the HNN exhibits stable states when $c_2 \in$ [5.5, 6.68]. When $c_2 \in$ [5.68, 5.95], the system shows periodic states, transient chaos and chaos switching with each other. As $c_2$ increases to 5.95, the HNN always exhibits periodic states.

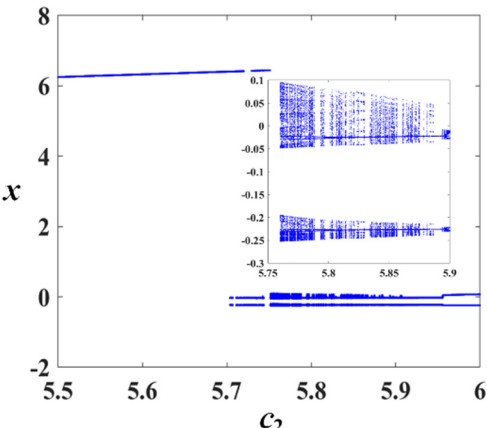

**Figure 12.** The bifurcation diagram of the HNN varying with crosstalk parameter $c_2$.

The HNN produces different attractors varying with the crosstalk parameter $c_2$, as shown in Figure 13. Notice that the transient chaos emerges in the HNN under $c_2$ = 5.68, as depicted in Figure 13a. In this case, the attractor is chaotic over a period of time, then switches into another nonchaotic behavior after the period of time. This phenomenon is a kind of special dynamics in neural networks, because it is difficult to find in a nonlinear system [23]. When solving the combinatorial optimization problem using the HNN, the introduction of transient chaos can help to jump from the local optimal solution to the global optimal solution. The HNN with transient chaos has stronger global search ability, so it has higher application values [24].

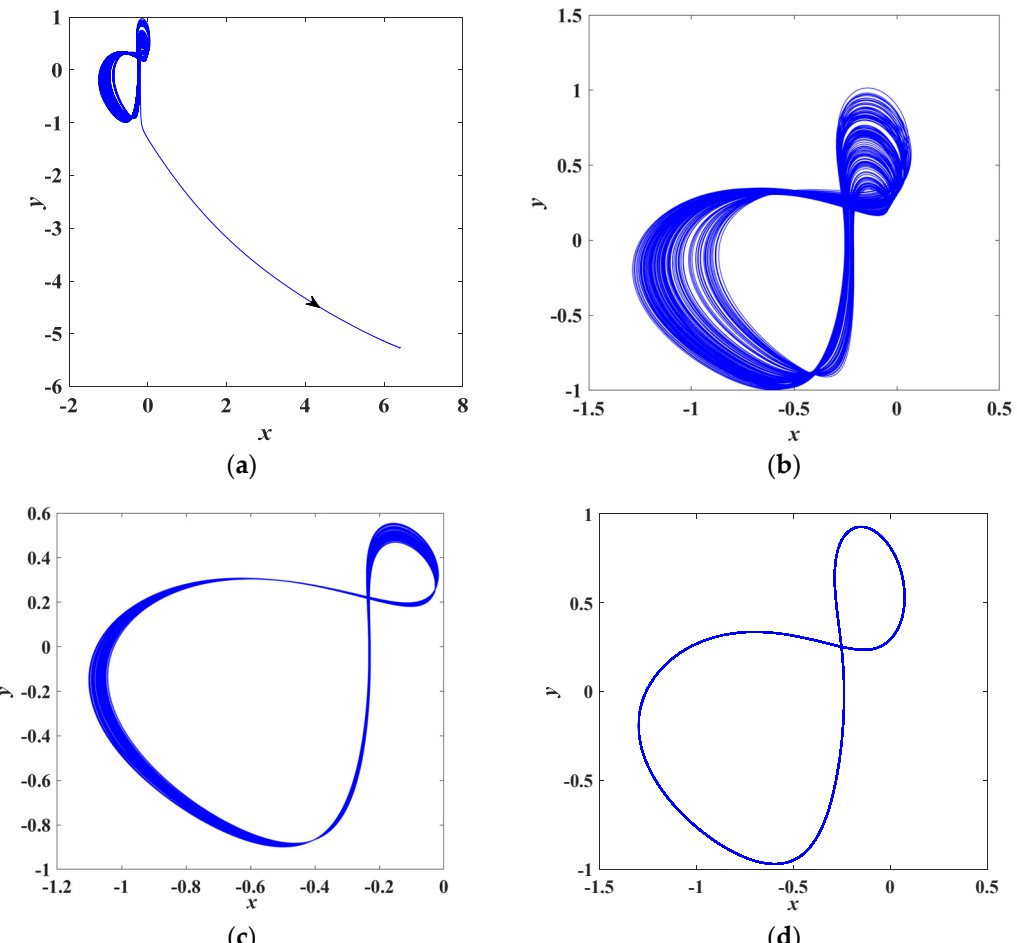

**Figure 13.** Phase portraits of the HNN under the parameter $c_2$ = 5.68, 5.76, 5.82, 6: (**a**) $c_2$ = 5.68; (**b**) $c_2$ = 5.76; (**c**) $c_2$ = 5.82; (**d**) $c_2$ = 6.

## 5. Sensitivity of Initial Conditions and Coexisting Behaviors

Chaos is sensitive to initial conditions, which is vividly described by the "butterfly effect" presented by Lorenz [25]. Small perturbations of initial conditions can eventually cause the separation of chaotic orbits. The sensitivity means the unpredictability of long-term nonlinear behaviors.

Coexisting phenomenon means different kinds of attractors emerge in a system when choosing the same system parameters and different initial values. If the obtained attractors have different dynamics, such as point attractors, periodic attractors, and chaotic attractors, these attractors are called inhomogeneous attractors. If the obtained attractors have the same dynamics but different gravity or shapes, these attractors are called homogenous attractors [26]. The generation of coexisting attractors indicates high sensitivity to initial conditions for the HNN, which also means rich dynamics.

Now, set the parameters $c_1$ = 5.55, $c_2$ = 5.9, and the initial value ($x(0)$, 0, $z(0)$, 0). The three-dimensional bifurcation diagram of the state $x$ varying with initial values $x(0)$ and $z(0)$ is depicted in Figure 14.

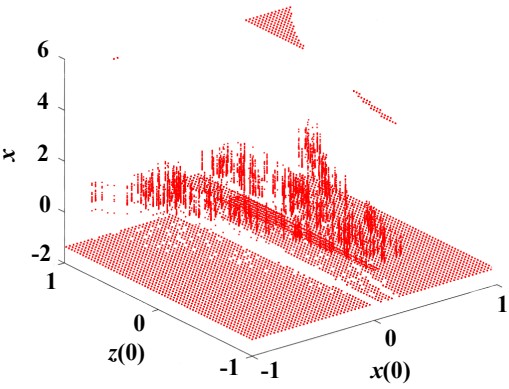

**Figure 14.** The three-dimensional bifurcation diagram of the HNN varying with $x(0)$ and $z(0)$.

Observe from Figure 14 that the chaotic and periodic orbits switch with each other over the range of $x(0) \in [-1,1]$ and $z(0) \in [-1,1]$, indicating that rich and complex dynamics emerge in the HNN.

When changing the initial values $x(0)$ and $z(0)$, the obtained basin of attraction and typical coexisting attractors on the *x-y-z* plane are shown in Figure 14.

Observe from Figure 15 that the two-neuron based HNN generates rich coexisting dynamics when changing initial values and fixing parameters, including the coexisting of periodic attractors and chaotic attractors, the coexisting of transient chaotic attractors and stable point attractors, the coexisting of chaotic attractors, periodic attractors and point attractors. The details are listed in Table 2.

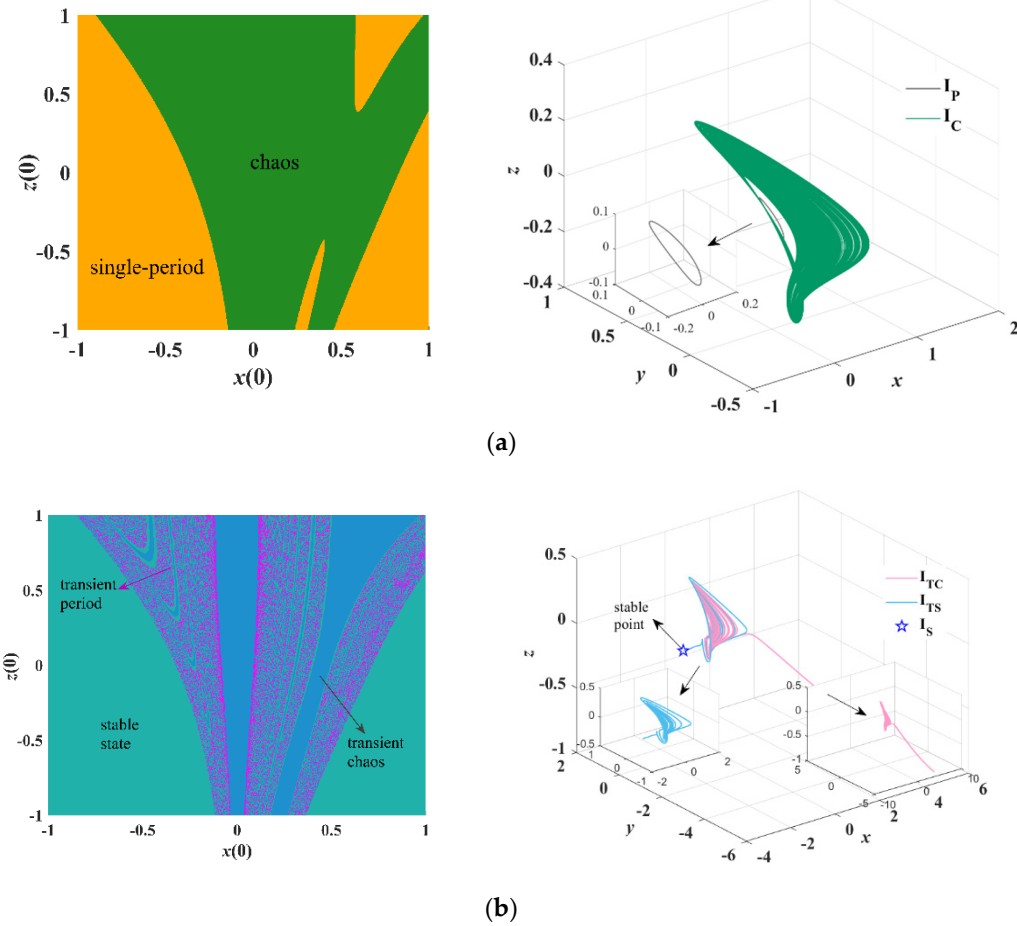

**Figure 15.** *Cont*.

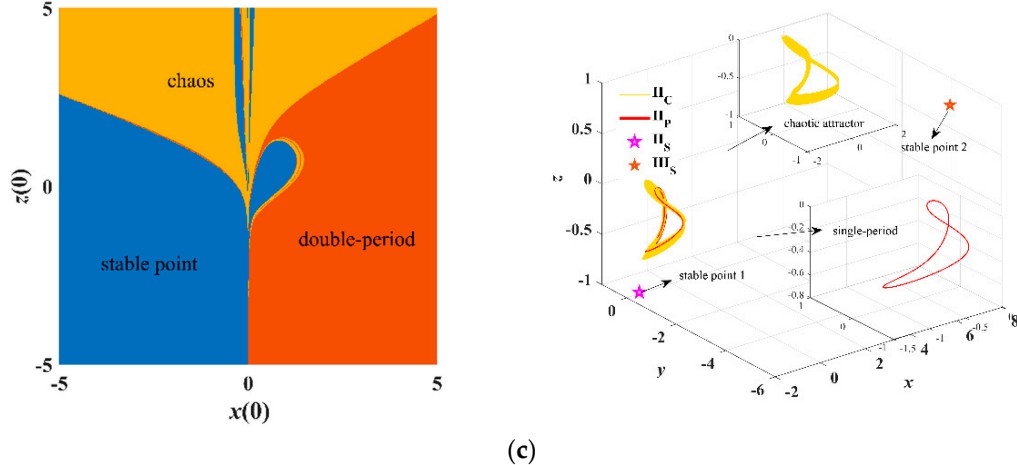

(**c**)

**Figure 15.** Basin of attraction and typical coexisting attractors under different parameters: (**a**) $a_2 = 1$, $c_1 = 5.1$, $c_2 = 3.2$; (**b**) $a_2 = 1$, $c_1 = 5.7$, $c_2 = 3.5$; (**c**) $a_2 = 1.04$, $c_1 = 5.55$, $c_2 = 5.9$.

**Table 2.** Characteristics of different attractors.

| Color | Characteristics | Types | Initial Values |
|---|---|---|---|
| | Single-period attractor | $I_P$ | $(-0.1, 0, 0.1, 0)$ |
| | Single-scroll chaos | $I_C$ | $(0.1, 0, -0.1, 0)$ |
| | Transient chaos | $I_{TC}$ | $(-0.2, 0, 0.2, 0)$ |
| | Transient periodic attractor | $I_{TS}$ | $(0.2, 0, -0.3, 0)$ |
| | Point attractor | $I_S$ | $(0.2, 0, -0.2, 0)$ |
| | Double-scroll chaos | $II_C$ | $(1, 0, -4, 0)$ |
| | Double-periodic attractor | $II_P$ | $(2.5, 0, 2, 0)$ |
| | Point attractor | $II_S$ | $(-1.5, 0, -1, 0)$ |
| | Point attractor | $III_S$ | $(1.5, 0, 0.5, 0, 0)$ |

## 6. Circuit Simulation

In order to verify the validity of the mathematical analyses, we made a circuit simulation by using the Pspice tool. Based on the HNN model described in Equation (6), we obtain

$$\begin{cases} RC_1 \frac{dv_x}{dt} = -v_x + \frac{R}{R_{11}}\tanh(v_x) - \tanh(v_y) \times \left[ \frac{R}{Ra_2} - \frac{R}{Rb_2}\tanh(v_u) + \frac{R}{Rc_2}\tanh(v_z) \right] \\ RC_2 \frac{dv_y}{dt} = -v_y + \tanh(v_x) \times \left[ \frac{R}{Ra_1} - \frac{R}{Rb_1}\tanh(v_z) + \frac{R}{Rc_1}\tanh(v_u) \right] + \frac{R}{R_{22}}\tanh(v_y) \\ RC_3 \frac{dv_z}{dt} = -v_z + \tanh(v_x) \\ RC_4 \frac{dv_u}{dt} = -v_u + \tanh(v_y) \end{cases} \quad (21)$$

where $v_x$, $v_y$, $v_z$ and $v_u$ represent the voltage of capacitors $C_1$, $C_2$, $C_3$ and $C_4$, respectively.

The main circuit of two-neuron based HNN from Equation (21) is shown in Figure 16a, including four ideal operational amplifiers, four "-tanh" function models, several resistors and capacitors. Figure 16b shows memristive synaptic equivalent circuits $W_1$ and $W_2$, including four multipliers and several resistors.

Here, set the time constant as 1 ms. The parameter values of circuit components are listed in Table 3. The obtained simulation results from Pspice software are depicted in Figure 17, which are consistent with the results obtained from MatLab.

In addition, we set $R_{11} = 10$ k$\Omega$ and $R_{22} = 5$ k$\Omega$. Different coexisting behaviors emerge in the circuit varying with the memristance $R_{a2}$. When $R_{a2} = 7.14$ k$\Omega$ and 6.67 k$\Omega$, the simulation results from Pspice are shown in Figure 18, which exhibit chaotic trajectory and

transient chaotic trajectory, respectively. The obtained results are consistent with that of Figure 10c,d.

In conclusion, the circuit simulation results verify the feasibility of the HNN circuit, which is conductive to the hardware implementation of neural networks and studying their synaptic crosstalk.

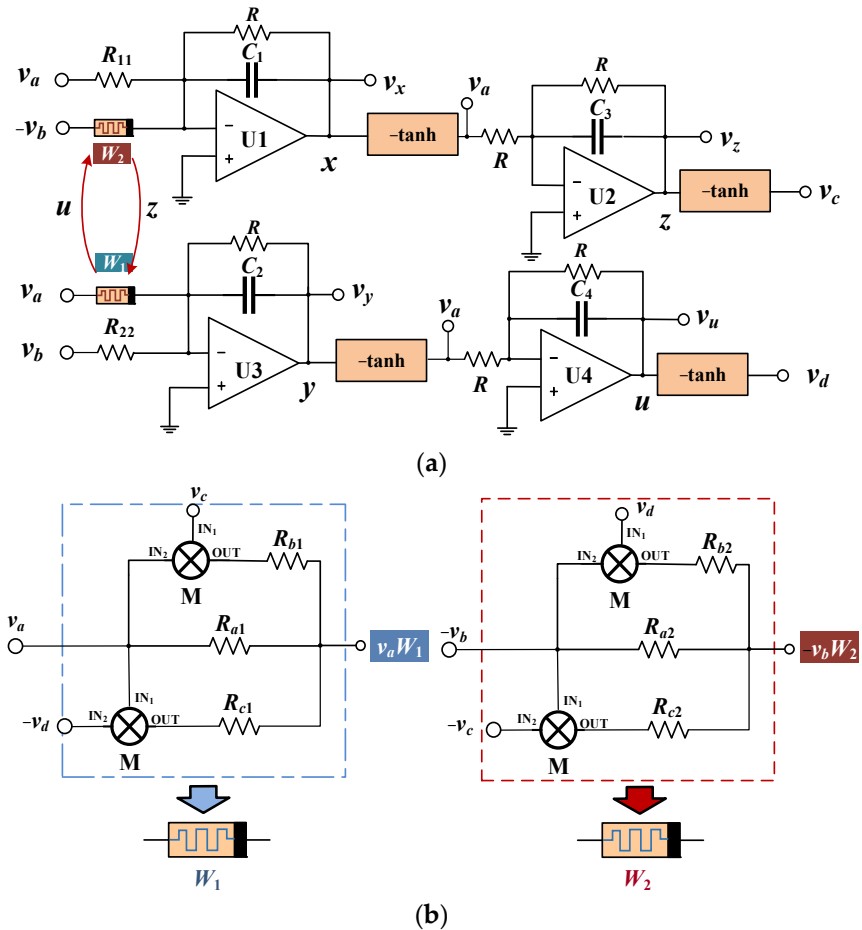

**Figure 16.** Circuit scheme of memristive HNN with synaptic crosstalk: (**a**) main circuit; (**b**) equivalent circuit of memristive synapse.

**Table 3.** Parameter values of components.

| Symbol | Parameter Values | Symbol | Parameter Values |
|--------|------------------|--------|------------------|
| $R$ | 10 kΩ | $R_{b1} = R/b_1$ | 250 kΩ |
| $C$ | 1 μF | $R_{a2} = R/a_2$ | 9.52 kΩ |
| $R_{11} = R/W_{11}$ | 8.06 kΩ | $R_{b2} = R/b_2$ | 333.33 kΩ |
| $R_{22} = R/W_{22}$ | 13.33 kΩ | $R_{c1} = R/c_1$ | 1.8 kΩ |
| $R_{a1} = R/a_1$ | 10 kΩ | $R_{c2} = R/c_2$ | 1.69 kΩ |

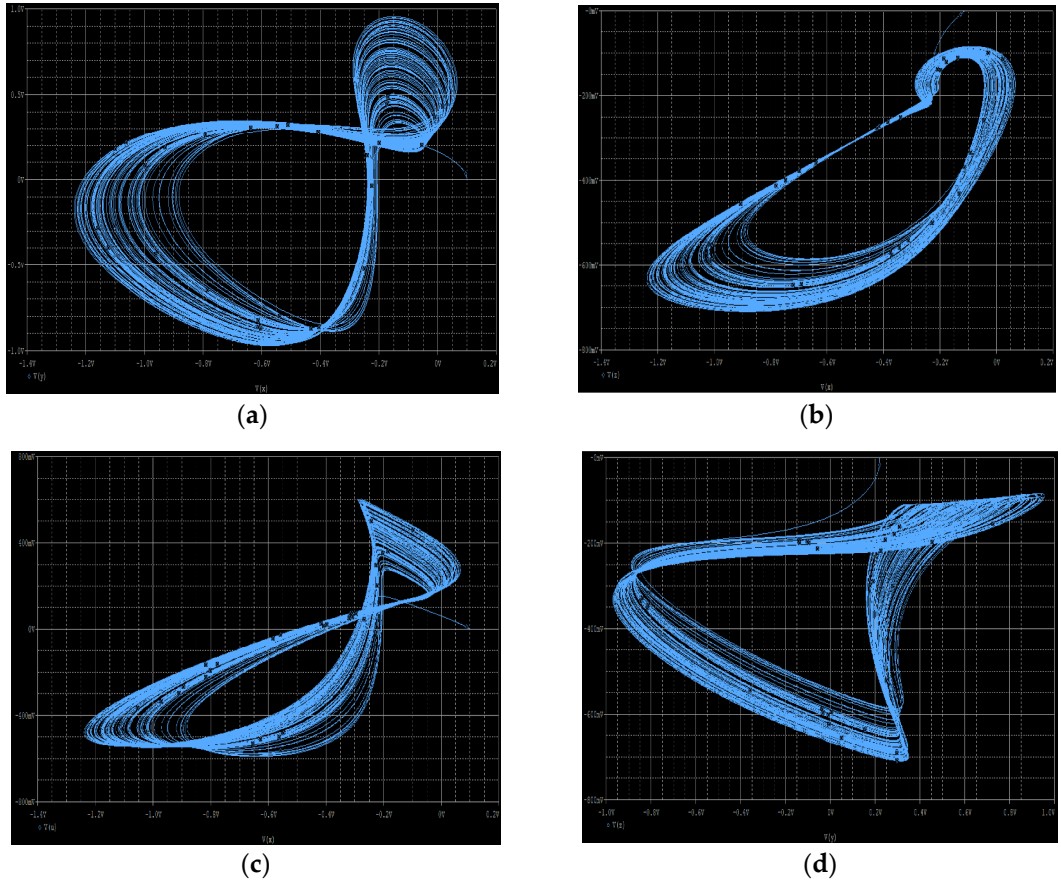

**Figure 17.** Simulation results of chaotic attractors: (**a**) *x-y* phase portrait; (**b**) *x-z* phase portrait; (**c**) *x-u* phase portrait; (**d**) *y-z* phase portrait.

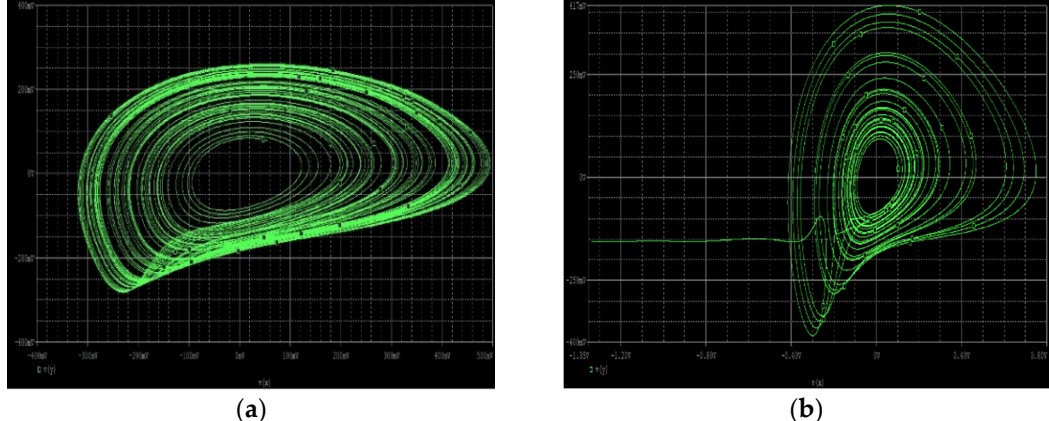

**Figure 18.** Simulation results of chaotic attractors and transient chaotic attractors: (**a**) $R_{a2}$ = 7.14 kΩ; (**b**) $R_{a2}$ = 6.67 kΩ.

## 7. Conclusions

Based on the synaptic plasticity and nonvolatility of the memristor, this paper presents a simple two-neuron-based Hopfield neural network, which can emulate the synaptic crosstalk of neural networks. By using the bifurcation diagram, basin of attraction and Lyapunov exponent spectrum, the dynamics of the HNN varying with memristive parameters and synaptic crosstalk weights are analyzed. Complex phenomena, including chaotic attractors, emerge in the HNN under the influence of synaptic crosstalk. In particular, a special phenomenon called transient chaos occurs in the HNN. Moreover, it is indicated

that the HNN has high sensitivity and rich coexisting dynamics via the phase portraits, bifurcation diagram and basin of attraction. Finally, the circuit simulation is completed via Pspice, which is consistent with the MatLab simulation results, further verifying the implementation of the hardware of memristive HNN.

**Author Contributions:** Conceptualization, R.Q., Y.D. and G.W.; methodology, R.Q. and X.J.; software, X.J.; validation, G.W. and Y.D.; formal analysis, Y.D.; investigation, G.W.; resources, R.Q.; writing— original draft preparation, X.J.; writing—review and editing, R.Q., G.W. and Y.D.; visualization, Y.D.; supervision, R.Q.; project administration, G.W.; funding acquisition, G.W. All authors have read and agreed to the published version of the manuscript.

**Funding:** This research was funded by the National Natural Science Foundation of China, grant number 61771176.

**Institutional Review Board Statement:** Not applicable.

**Informed Consent Statement:** Not applicable.

**Data Availability Statement:** Not applicable.

**Conflicts of Interest:** The authors declare no conflict of interest.

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
