# Peer review of "Two-Neuron Based Memristive Hopfield Neural Network with Synaptic Crosstalk"

_electronics, doi:10.3390/electronics11193034_

Round 1

Reviewer 1 Report

Dear Authors,

Please make the following corrections:

1. Line 31 - Explain "plasticity of synaptic weights"

2. Line 33 - "can flexibly solve different kinds of problems". Please give examples of these problems.

3. Please provide a practical application of your neural network.

4. Please summarize how your network differs from those described in other publications. What's the novelty of your approach?

5. The number of references to publications is definitely insufficient. Please complete the publication with more aspects of neural networks, their principles of operation, applications, advantages and disadvantages, optimization methods in relation to your neural network.

Several publications on this topic that may be useful and which describe the issues of neural networks or optimization methods are:

https://doi.org/10.3390/app11135771

https://doi.org/10.3390/pr9081292

https://doi.org/10.3390/cryst11111371

https://doi.org/10.3390/s22030813

https://doi.org/10.3390/math7111133

https://doi.org/10.3390/s21134567

6. Why did the authors use a hiperbolic tangent activation function? Please explain your choice. Whether range is the only selection criterion. What about effectiveness?

Reviewer 2 Report

The authors proposed a two-neuron-based memristive Hopfield neural network with a hyperbolic memristor emulating synaptic cross-talk. The analysis is performed using phase portraits, bifurcation diagrams, Lyapunov exponents, and basin attraction. The circuit implementation showed that the attractor plots matched those of the simulation results.

The article is interesting, and it merits publication. But I have a few recommendations, which I firmly believe are necessary to include in the manuscript. My comments include

1.     On page 2, explain Figure 1 in terms of what each component is doing. Also, provide the component types used, such as for Opamp and transistors. Also include the resistor values. Though it is on Page 14, I believe the reader needs to wait till the end to see them.

2.     On page three, line 76, what are a and b and what are their values

3.     Again, explain Figure 3 just like mentioned above (Figure 1). What is the capacitor in the feedback doing?

4.     Also, from Figure 4, make a note of how this is different from traditional synchronization, as discovered by Pecora, Louis M., and Thomas L. Carroll. "Synchronization in chaotic systems." Physical review letters 64.8 (1990): 821.

5.     In Figure 5, make a note of what W11 and W22 are.

6.     The Lyapunov function analysis is well done.

7.   What are the bars over (.) in line 154 indicate?

8.     In Table 1, how are equilibrium points obtained? (All four equilibrium points). An explanation is required as Figure 6 shows only two sets of equilibrium points. Also, in row 4, since we have a positive Eigenvalue (2.3973), isn't the system unstable? Correction is necessary here. 

9.     Explain Figure 7, such as what each attractor plot is showing.

10. While computing Lyapunov exponents, the authors mentioned they used the Jacobian matrix. I am puzzled by that. Did they use the QR decomposition method, which involved the Jacobian matrix? Please elaborate on this.

11. In line 188, the authors mentioned problems. What are those problems?

12.  Figure 9 looks good. But I recommend using a subfigure with two rows and 1 column. In Matlab, it is the subplot(21_). Use the same x-axis limits. That way, we can visualize how the dynamics are evolving and, consequently, the Lyapunov exponent spectrum.

13.  In Figure 10, you may want to use the type of dynamics for specific parameters. For example, a2 = 1.1, 1-period, etc.

14. In Line 232, a proper reason should be given. For instance, "This phenomenon is a kind of special into dynamics in neural networks." Similarly, Line 235, "The HNN with transient chaos has stronger global search ability, so it has higher application values."

15.  Figure 14(a) is not necessary. Because a positive value of the largest Lyapunov exponent already indicates it is sensitive to initial conditions. So this is not conveying any information to the reader.

16. I am concerned with Figures 14(b) and (c). The histograms of a chaotic waveform should have an invariant density function. Even though you change the initial conditions, the histogram shape should be the same. More information on this can be found in Flores, Benjamin C., Emmanuel A. Solis, and Gabriel Thomas. "Assessment of chaos-based FM signals for range–Doppler imaging." IEE Proceedings-Radar, Sonar and Navigation 150.4 (2003): 313-322.

I think the authors may not have eliminated the transients of a chaotic system. That is why simulations show different histograms in Figures 14(b) and (c). The authors need to recheck this.

17. What is 'x' in Figure 15?

18.  I strongly recommend authors tabulate the component values mentioned in lines 309 to 312. In addition, they should also equate each parameter value. For example, c1 = R/ ___, c2 = R/___ , etc.  

19.  In conclusion, I recommend removing histograms (337) unless there is compelling evidence and rigorous mathematical explanation.

Round 2

Reviewer 1 Report

Authors made the suggested corrections. Now, the article can be considered suitable for publication.